# Neural Topological Ordering for Computation Graphs

**Mukul Gagrani**[*]
Qualcomm AI Research[†]
mgagrani@qti.qualcomm.com

**Corrado Rainone**[*]
Qualcomm AI Research
crainone@qti.qualcomm.com

**Yang Yang**[‡]
Google LLC

**Harris Teague**
Qualcomm AI Research

**Wonseok Jeon**
Qualcomm AI Research

**Herke Van Hoof**
University of Amsterdam, Netherlands

**Weiliang Will Zeng**
Qualcomm AI Research

**Piero Zappi**
Qualcomm AI Research

**Christopher Lott**
Qualcomm AI Research

**Roberto Bondesan**
Qualcomm AI Research

## Abstract

Recent works on machine learning for combinatorial optimization have shown that learning based approaches can outperform heuristic methods in terms of speed and performance. In this paper, we consider the problem of finding an optimal topological order on a directed acyclic graph with focus on the memory minimization problem which arises in compilers. We propose an end-to-end machine learning based approach for topological ordering using an encoder-decoder framework. Our encoder is a novel attention based graph neural network architecture called *Topoformer* which uses different topological transforms of a DAG for message passing. The node embeddings produced by the encoder are converted into node priorities which are used by the decoder to generate a probability distribution over topological orders. We train our model on a dataset of synthetically generated graphs called layered graphs. We show that our model outperforms, or is on-par, with several topological ordering baselines while being significantly faster on synthetic graphs with up to 2k nodes. We also train and test our model on a set of real-world computation graphs, showing performance improvements.

## 1 Introduction

Many problems in computer science amount to finding the best sequence of objects consistent with some precedence constraints. An intuitive example comes from routing problems, where we would like to find the shortest route between cities but we have requirements (i.e. for example to pick up and subsequently deliver a package) on the order in which the cities should be visited [1]. Another case is found in compiler pipelines, wherein the "cities" become operations to be executed and the constraints come from the data dependencies between these operations, such as when the result of an operation is an operand in a subsequent one. In this case, the metric to be optimized can be the run time of the compiled program, or the memory required to execute the program [2]. Common across this class of problems is their formulation in term of finding the optimal topological order

---

[*]Equal contribution

[†]Qualcomm AI Research is an initiative of Qualcomm Technologies, Inc.

[‡]Work completed during employment at Qualcomm Technologies, Inc.

36th Conference on Neural Information Processing Systems (NeurIPS 2022).

of the Directed Acyclic Graph (DAG) that encodes the precedence constraints, which induces a Combinatorial Optimization [3] (CO) problem which is in general computationally hard [4].

Already from the two examples above, one can immediately grasp the relevance of such problems for industrial Operations Research, which has prompted various actors to invest in the development of efficient CO solvers; these solvers usually encapsulate heuristic methods whose design typically requires extensive use of domain-specific and problem-specific knowledge, across decades of development. In recent years, considerable interest has emerged in the possibility of replacing such handcrafted heuristics with ones learned by deep neural nets [5] (machine learning for combinatorial optimization, MLCO). As a matter of fact, both of our two examples of DAG-based CO problems have indirectly been object of study in the Machine Learning literature. References [6, 7, 8, 9] take into consideration Routing Problems, especially the Traveling Salesperson Problem (TSP) which, on account of its richness, complexity and long history of mathematical study [10], has attained the status of a standard benchmark for MLCO [8]. Conversely, less attention has been devoted to operations sequencing likely due to the proprietary and sensitive nature of compiler workflows, which hampers the definition of public benchmarks. References [11, 12] both consider the task of optimizing the run time of a neural network's forward pass by optimizing the ordering and device assignment of its required operations. However, in this last case the sequencing stage is only one part of a larger compiler pipeline, and as a result of this both the performance metrics and the datasets employed cannot be made available for reproduction by third parties. This makes it both hard to assess the results therein, and to draw general conclusions and guidelines for the advancement of MLCO, which still suffers from a lack of commonly accepted and standard datasets and benchmarks.

In this work, we address the problem of finding optimal topological orders in a DAG using deep learning, focusing on the compiler task of optimizing the peak local memory usage during execution. We make the following contributions:

- We present a neural framework to optimize sequences on directed acyclic graphs. Mindful of the need for scalability, we consider a non-auto-regressive (NAR) scheme for parametrizing the probability distribution of topological orders. This allows our method to attain an extremely favorable performance vs. run time tradeoff: it always outperforms fast baselines, and is only matched or outperformed by those requiring a much longer (in one case 4000x more) run time.
- We address the problem of how to perform meaningful message-passing on DAGs, a graph type which has received comparatively less attention in the literature on Graph Neural Networks. We introduce *Topoformer*, a flexible, attention-based architecture wherein messages can be passed between each and every pair of nodes, with a different set of learnable parameters depending on the topological relation between the nodes.
- To test our method, we introduce an algorithm for the generation of *synthetic*, layered, Neural Net-like computation graphs, allowing any researcher to generate a dataset of *as many as desired* graphs of *any desired size*. These graphs are a more faithful model of real NN workflows, and allow us to prove our method on a much larger and varied dataset, than previous efforts [11]. To our knowledge, this is the first public algorithm of this kind. Nevertheless, we also test our method on proprietary graphs to illustrate its relevance to realistic compiler workflows.

## 2   Related work

**Machine Learning for Combinatorial Optimization:**   Combinatorial optimization as a use case for deep learning poses interesting technical challenges. First, the combinatorial nature of the problem conflicts with the differentiable structure of modern deep neural networks; and second, the models need to be run at large scale to solve real world instances, exacerbating the challenges in training deep learning models. Given the discrete nature of CO problems, a natural approach is to pose them as reinforcement learning (RL) problems [13]. The aim is then to learn a policy that selects the best actions to maximize a reward directly related to the optimization objective. Algorithms then differ in the way the policy is parameterized: either in an end-to-end manner where the actions directly correspond to solutions of the optimization problem [12, 6, 8, 14], or in a hybrid manner, where the policy augments parts of a traditional solver, e.g. by replacing heuristics used in setting parameters of an algorithm, see e.g. [11, 9, 7, 2]. Our approach follows an end-to-end design philosophy, which, not having to rely on an external algorithm, affords better control of post-compile run time and facilitates application on edge devices [2]. Furthermore, RL has the advantage of being useful as a black box optimizer, when no handcrafted heuristics can be designed.

**Sequence Optimization via ML:** Within MLCO, much effort has been devoted to the task of predicting optimal sequences [15, 6, 16, 17, 18]. The end-to-end nature of our method places it close to the one proposed in [6], although to the best of our knowledge, our work is the first to tackle the challenge of enforcing precedence constraints in the network predictions. As we shall see in more detail below, this generalization is non-trivial: already counting the number of topological orders belongs to the hardest class of computational problems [4]. This has to be contrasted with the fact that the number of sequences without topological constraints is simply $n!$ for $n$ objects. Besides, as pointed out in [8], no MLCO method has so far been able to convincingly tackle TSPs of sizes above a few hundred nodes, when it comes to *zero-shot* generalization to unseen problem instances, i.e. when no fine tuning on the test set is done. It is also therein pointed out how an auto-regressive parametrization of the sequence (which was the method used in ref. [6]) appears to be necessary to achieve acceptable performance even at those small sizes. Conversely, in the present work we show compelling zero-shot performance on DAGs of sizes up to *thousands* of nodes, while nonetheless generating our sequences in a fully non-auto-regressive (NAR) way and maintaining a strong run time advantage over classical sequencing algorithms. Our results can then also be interpreted as cautioning against the idea of using the TSP as the sole, paradigmatic test-bed for MLCO research, as [8] remarks.

**ML for Compiler Optimization:** The DAG sequencing task we consider is an omnipresent stage in compiler workflows, which usually also include such tasks as device assignment and operations fusion [12]. In such a setting, jointly optimizing these tasks to reduce the *run time* of a certain workflow (such as the forward pass of a Neural Net) is a common objective, which in refs [11, 12, 19] is tackled with ML methods. In this work we focus on the task of minimizing the peak local memory usage during execution, which does not require a performance model or simulator as well as being relevant to applications on edge devices [2]. In [11], the ML solution leans on an existing genetic algorithm, whilst our solution is end-to-end, much like that proposed in [12]. Another characteristic of the solution proposed in [12] is the idea of interpolating between AR and NAR via an *iterative refinement* scheme, in which sequences are generated in one pass but subsequently refined during an user-defined number of subsequent passes; conversely, we generate all our sequences in a single pass. While in [11] the run time optimization is studied on both real-world and synthetic random graphs – the latter being relatively small (up to about 200 nodes), the peak memory optimization is studied only on a proprietary dataset augmented via perturbation of the node attributes. In [12] the authors train and test their method on a relatively small set of six proprietary workflows which are not disclosed to the reader, and out of those six, only the size of the largest instance is mentioned.

**Deep Graph Neural Networks:** Given that our problem is specified as a DAG, it is a logical choice to parametrize our sequence-generation policy with a Graph Neural Network architecture [20]. The basic idea of every GNN architecture is to update graph and edge representations by passing messages between the graph nodes along the graph edges [21]. However, this can be too restrictive when it comes to sequence generation on DAGs. For example, nodes that come after each other in the sequence might not be linked by an edge in the graph, and therefore are unable to directly influence each other's representation. Notice how this difficulty is another consequence of the presence of precedence constraints in our problem, which conversely was not an issue in e.g. [6] where the graph is fully connected and no constraints are present. Relatively few efforts (see e.g. [22, 23, 24]) have been devoted to devise a way to perform meaningful message passing on DAGs. As a matter of fact, the quest for expressive GNN architectures is at the center of intense theoretical investigation [25, 26].

## 3 Background

### 3.1 Topological Orders and DAGs

We here introduce the mathematical background, starting with a few definitions. A partial order is an irreflexive transitive relation $<$ between certain pairs of a set $V$. We call a pair $(x, y) \in V \times V$ that is related by $<$ comparable, and *incomparable* otherwise. A Directed Acyclic Graph (DAG) $G = (V, E)$ is a directed graph with no directed loops. We can map a DAG $G = (V, E)$ to a partially ordered set $(V, <)$ where $x < y$ if there is a directed path from node $x$ to node $y$. Multiple DAGs map to the same partial order. For example, the DAGs with vertex set $\{x, y, z\}$ and edge sets $E = \{x \to y, y \to z\}$ and $E' = \{x \to y, y \to z, x \to z\}$, where $s \to t$ denotes a directed edge from $s$ to $t$, correspond to the same partial order $x < y < z$. We define the *transitive closure* (TC) of a DAG as the graph with most edges that has the same underlying partial order, so that there exists a directed edge $(x, y)$

whenever $x < y$. Conversely, the *transitive reduction* (TR) is the graph with *least* edges that results in the same partial order. We denote the order induced by a DAG by $<_G$.

A topological order or sorting of a DAG $G$ is a bijection $\sigma : V \to \{1, \ldots, |V|\}$ such that $\sigma(x) < \sigma(y)$ whenever $x <_G y$. The set $\mathcal{T}_G$ of topological orders of $G$ is a subset of the permutation group of the vertices and coincides with total orders on $V$ that respect $<_G$, called *linear extensions* of the partial order. While there are several well-known algorithms to compute a topological order of a DAG, e.g. breadth first search and depth first search, counting the number of topological orders is one of the hardest computational problems, being #P complete [4]. In this work we develop a general machine learning method to find a topological order that minimizes a given cost function on a DAG, which we define in the next section.

## 3.2 Peak Memory Minimization

Deciding the best way to schedule operations in a computational graph representing a neural network is a central problem in compilers [11, 12, 2]. We can associate a DAG to a computational graph in such a way that nodes represent operations ("ops"), and incoming/outgoing edges represent operands/results of these operations. Every time one executes an op, the inputs[4] to that op need to be in memory, and memory for the outputs needs to be allocated. Therefore, each node of the DAG carries a label $m : V \to \mathbb{N}$ specifying the memory required to store the output of that op. A typical first step in scheduling a DAG is to identify topological orders to execute operations. Compilers for edge devices, which have limited memory, aim at choosing the optimal topological order that minimizes the peak memory footprint [2]. We focus therefore on the peak local memory usage minimization task, which can be formulated as the following combinatorial optimization problem on a labeled DAG $G = (V, E, m)$:

$$\min_{\sigma \in \mathcal{T}_G} \mathcal{C}(\sigma), \qquad \mathcal{C}(\sigma) = \max(M_1(\sigma), \ldots, M_{|V|}(\sigma)), \tag{1}$$

with the definitions

$$M_t = I_{t-1} + m(\sigma_t), \tag{2}$$

$$I_t = M_t - \sum_{i \in S_t} m_i, \qquad S_t = \left\{ i : i \notin \bigcup_{l=0}^{t-1} S_l \text{ and } \forall (i,j) \in E, j \in \sigma_{1:t} \right\}, \tag{3}$$

i.e. the memory usage at time $t$ is given by the memory usage $I_{t-1}$ of the outputs which have not yet been consumed, at time $t-1$, by downstream operations, plus the memory requirement of the output of operation $\sigma(t)$. $I_t$ is in turn obtained by subtracting from $M_t$ the memory costs of nodes whose outgoing edges only connect to already scheduled nodes, i.e. nodes whose output was only required by already scheduled operations. Naturally, $I_0 = 0, S_0 = \phi$.

## 4 Method

We use an encoder-decoder architecture whose schematic is shown in figure 1. Our encoder is *Topoformer*, a novel GNN architecture, which derives an embedding for each node of the graph. The embeddings are used by the decoder which generates a distribution in the sequence space and finally the distribution can be converted to a sequence via different inference methods like sampling, greedy inference or beam search. Next, we describe each of the component in detail.

### 4.1 Topoformer: Topologically Masked Attention

A Graph Neural Network (GNN) is a natural choice to encode our scheduling problem via embedding of the DAG nodes. All canonical GNN architectures operate by updating these embeddings via the aggregation of "messages" sent from the other nodes, usually in the form of some function of their own embedding [20]. Architectures mainly differ in how the set of sender nodes is constructed and the aggregation function is chosen. In a Graph Convolutional Network [27], the senders are the first neighbors of a node and the aggregation function is a weighted average, whilst in a vanilla Graph Attention Network [28], the senders are all the other nodes, but their contributions are aggregated

---

[4]We use "inputs" and "operands" interchangeably throughout the paper.

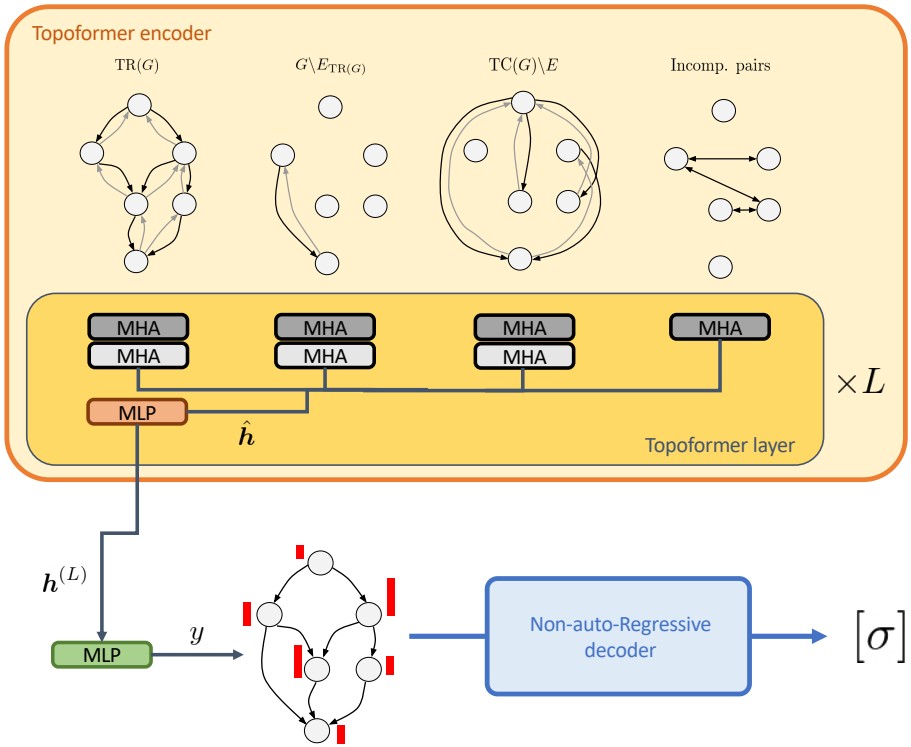

Figure 1: Our complete architecture for neural topological ordering. The shades of gray in the MHA boxes are to highlight how attentions heads operate separately on the forward and backward version of the first three graphs. The priorities $(y_i)_{i=1}^{|V|}$ are represented by the red bars on the original DAG and decoded into a sequence with its associated probability.

via averaging with *learned* weights so as to account for their degree of relevance. When trying to apply such mechanisms on DAGs, a common point of contention is whether, and how in practice, the partial ordering encoded by it should reflect in the direction of travel of the messages [29, 24, 22]. While disregarding the DAG structure entirely (as one would do in a vanilla GAT), does not appear wise, it might be too restrictive when it comes to our task. For example, nodes that are next to each other in the sequence might well be incomparable, and thus lack a path for messages between them. The combinatorial nature of the task also poses requirements; it is known [25, 5] that reasoning about CO problems on a graph requires the capacity to reason about the *global structure* of it, whilst architectures such as those proposed in [29, 24, 22] limit the set of sender nodes to a *local* neighborhood of the receiver node. In summary, our architecture must strike a compromise between accounting for *global* structure and *local* partial ordering information.

Our *Topoformer* architecture meets these requirements. A vector $x_i$ of input features (see the appendix for details about its definition and dimensionality) is first turned into an initial node embedding $h_i^{(0)}$ via a node-wise linear transformation, $h_i^{(0)} = W x_i + b$. Subsequently, a succession of $L$ attention layers, each of them consisting of a Multi-Head Attention (MHA) [28] sub-layer followed by one more node-wise MLP, updates these embeddings, similar to a vanilla Transformer [30]; however, we confer a topological inductive bias to these updates by having a separate group of attention heads masked by each of the following graphs induced by the original DAG:

- Its transitive reduction (TR).
- The directed graph obtained by removing the TR edges from the DAG: $G \backslash E_{\mathrm{TR}(G)}$.
- The directed graph obtained by removing the edges of the DAG from its TC: $\mathrm{TC}(G) \backslash E$.
- The backwards versions (i.e. with flipped edges) of each of the three above.
- The undirected graph obtained by joining all incomparable node pairs.

By adding together these graphs, one would obtain the fully connected graph relative to the node set $V$, whereupon all nodes would attend to all nodes. Then effectively, the propagation rules of Topoformer are same as those of a vanilla transformer encoder,

$$\hat{\boldsymbol{h}}_i^{(\ell)} = \boldsymbol{h}_i^{(\ell-1)} + \texttt{concat}_j \left[ \text{MHA}_i^{\ell,j} \left( \boldsymbol{h}_1^{(\ell-1)}, \ldots, \boldsymbol{h}_{|V|}^{(\ell-1)}; M^j \right) \right], \tag{4}$$

$$\boldsymbol{h}_i^{(\ell)} = \hat{\boldsymbol{h}}_i^{(\ell)} + \text{MLP}^{(\ell)} \left( \hat{\boldsymbol{h}}_i^{(\ell)} \right), \tag{5}$$

save for the presence of the *mask* $M^j$, which ensures that head $j$ only attends to its assigned graph among the seven listed above. Following [31], we also apply layer normalization [32] to the MHA and MLP inputs. The number of heads assigned to each graph can be chosen independently (setting it to zero means to not message-pass along the edges of the respective graph), or parameters can be tied among different MHAs. One should also remark how the MLP sub-layer allows the flow of information between different attention heads. All nodes are then able to influence each other's representation, while anyway injecting a strong inductive bias based on the DAG structure. Information about the Topoformer configurations used in our experiments is provided in the appendix.

## 4.2 Decoder

Once the embeddings of the nodes are generated, the decoder's task is to derive a stochastic policy $p(\sigma|G)$ over the valid topological orders of the graph. The most straightforward way is to take advantage of the chain rule of conditional probability to decompose the policy as a product

$$p(\sigma|G) = \prod_{t=2}^{|V|} p_\theta(\sigma_t|\sigma_{1:t-1}, \boldsymbol{h}, G) \times p_\theta(\sigma_1|\boldsymbol{h}, G), \tag{6}$$

We could then sample a complete sequence by autoregressively choosing a new node at each step as done e.g. in [6]. This scheme is the most principled and expressive; however, when a NN is used as a function approximator for $p_\theta$, it also requires that $|V|$ calls to this NN be performed, which limits its feasibility to relatively small graphs due to the amount of computation required.

In order to scale to large graphs, we employ a Non-Auto-Regressive (NAR) scheme which decouples the number of NN calls from the graph size. Similar to the approach of [12], we assign scheduling *priorities* $y_i \in \mathbb{R}$ to the nodes, rather than scheduling probabilities. The priority for node $i$ is derived by passing its final embedding through an MLP:

$$y_i = \text{MLP} \left( \boldsymbol{h}_i^{(L)} \right). \tag{7}$$

These priorities are assigned with a *single* NN inference. The sequence itself is subsequently constructed by adding a new node at each step. Given the partial sequence $\sigma_{1:i-1}$, the next node can only be selected from a subset $\mathcal{S}(\sigma_{1:i-1}, G)$ of schedulable nodes, due to both the graph topology and choices made earlier in the sequence. Then, the distribution of the next node to be added at step $i$ is given as follows:

$$p(\sigma_t|\sigma_{1:t-1}, \boldsymbol{h}, G) = \begin{cases} \dfrac{\exp(y_{\sigma_t})}{\sum_{j \in \mathcal{S}(\sigma_{1:t-1}, G)} \exp(y_j)}, & \text{if } \sigma_t \in \mathcal{S}(\sigma_{1:t-1}, G), \\ 0, & \text{otherwise.} \end{cases} \tag{8}$$

**Decoding Methods**: We use the following three methods to obtain the next node in the partial sequence from the distribution $p(\sigma_t|\sigma_{1:t-1}, \boldsymbol{h}, G)$:

1. *Greedy*: At each step $t$, select the node with the highest probability i.e. $\sigma_t = \arg\max_{\tilde{\sigma}_t} p(\tilde{\sigma}_t|\sigma_{1:t-1}, \boldsymbol{h}, G)$
2. *Sampling*: At each step $t$, sample from the next node distribution i.e. $\sigma_t \sim p(\cdot|\sigma_{1:t-1}, \boldsymbol{h}, G)$
3. *Beam search with state-collapsing*: We can also expand the partial sequences by using a beam search method where the score function is total probability of the partial sequence. We improve our beam search routine by making the following observation: suppose there are two partial sequences in consideration, $\sigma_{1:t}$ and $\tilde{\sigma}_{1:t}$, such that both have scheduled the same set of nodes so far (but different order), and $\mathcal{C}(\sigma_{1:t}) < \mathcal{C}(\tilde{\sigma}_{1:t})$. Then, we can ignore the partial sequence $\tilde{\sigma}_{1:t}$ and only keep $\sigma_{1:t}$ in the beam search. This is because both partial sequences must schedule the same set of remaining nodes, and hence the set of future memory costs are identical for both $\sigma_{1:t}$ and $\tilde{\sigma}_{1:t}$, but the current peak memory cost is higher for $\tilde{\sigma}_{1:t}$. Thus, $\sigma_{1:t}$ dominates $\tilde{\sigma}_{1:t}$ in terms of achievable minimal peak memory usage.

### 4.3 Training

Our encoder-decoder architecture induces a distribution $p_\theta(\sigma|G)$ on the set of topological orders for a given DAG $G$. The expected cost incurred is given by $J(\theta|G) = \mathbb{E}_{p_\theta(\sigma|G)}[\mathcal{C}(\sigma(\theta))]$. We minimize the cost $J(\theta) = \mathbb{E}_G[J(\theta|G)]$ via gradient descent using the REINFORCE gradient estimator [33, 13] as follows

$$\nabla J(\theta) = \mathbb{E}_{G,p_\theta(\sigma|G)}[(\mathcal{C}(\sigma) - b(G))\nabla_\theta \log p_\theta(\sigma|G)], \tag{9}$$

where $b(G)$ is a *baseline* meant to reduce the variance of the estimator. We follow [6] in setting it equal to the cost of a *greedy rollout* of a baseline policy on the graph $G$

$$b(G) = \mathcal{C}(\arg\max_\sigma p_\theta(\sigma|G)). \tag{10}$$

## 5 Experiments

We conduct experiments on a synthetic dataset of graphs which we refer to as "layered graphs", as well as a set of real-world computation graphs. We compare our approach with the following classic topological ordering baselines:

- *Depth/Breadth first sequencing*: Find the topological order by traversing the graph in depth/breadth first manner according to the layout of the graph generated using pygraphviz.
- *Depth-first dynamic programming (DP)*: Depth-first DP is a global depth-first method for searching the optimal sequence, with automatic backtracking when equivalent partial sequences are found; it retains the full sequence with minimum cost so far, and returns it if search does not complete before the prescribed timeout.
- *Approximate DP*: In approximate DP, a beam of partial sequences are considered at each step. For each beam in the subsequent step, the top-$K$ partial sequences with the lowest costs are retained. This DP is also able to find the optimal sequence given enough memory and compute resources (with unlimited beam size $K$), but we only consider an approximate version with $K = 10^5$ in this work. Note that approximate DP uses the state-collapsing and parallelism to improve its efficiency.
- *Random order*: We generate 100 random topological orders, and pick the one with smallest cost.

Please see the appendix for more detailed description of the baselines. Therein, we also report some ablation studies on various neural baselines including ablation studies on decoder by considering other neural architectures, as well as a comparison with an end to end baseline adapted from ref. [6]. Neural topo order greedy, sample and BS denote the performance of our model in greedy, sampling and beam search inference mode respectively. We use a sample size and beam size of 16 sequences, of which the best one is subsequently picked, for all our experiments. Next, we describe in detail the results of the two experiments.

### 5.1 Layered Graphs

In order to generate a large corpus of training data we come up with a way to synthetically generate graphs of a given size which have similar structure to the computation graphs of feed-forward neural networks. We call our synthetic graph family *layered graphs*, as these graphs comprise of well-defined layers of nodes. The nodes in a layer have connections to the nodes in the subsequent layer and can also have skip connections with nodes in layers farther down. The number of layers, number of node per layer, number of edges between subsequent layers, number of skip connections and memory utilization of the nodes are all generated randomly, and can be controlled by setting appropriate parameters. We refer the reader to the appendix for more details on layered graphs, including their generation algorithm and some visual examples.

We train our model on 500-node layered graphs for 325 epochs, where in each epoch we generate a training set of 1000 new graphs. We test the performance of our model on a set of 300 unseen graphs of the same size, generated with the same method. We also evaluate the cross-size generalization performance of our trained model by testing it on graphs of size 1000 and 2000. We refer the reader to the appendix for a comprehensive description of the training algorithm and model configuration.

Figure 2 shows the performance vs. run time plot on layered graphs of size $|V| = 500, 1000$, and 2000. We report the performance in terms of the % gap of peak memory utilization from the peak memory obtained via approximate DP, which we consistently observed to be the the best-performing

baseline. Note that the run time is plotted on a log-scale. We can observe that for 500-node graphs, our model beats all the baselines except approximate DP in terms of both the memory usage and run time. Our model is slightly worse than approximate DP from the memory usage perspective, but it runs 100x faster. We also observe that our model generalizes well to larger sized graphs. For the case of 2000-node graphs our model performs better than approximate DP in terms of peak memory usage, while being 4000x times faster. This shows that while approximate DP performs more poorly as graph size increases, our model is able to generalize to larger graphs by learning meaningful embeddings of the topological structure thanks to Topoformer, and to be extremely fast thanks to our NAR decoding scheme.

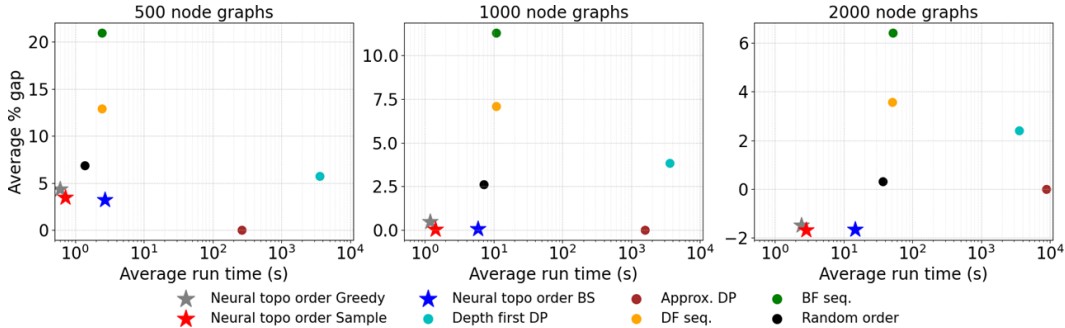

Figure 2: Average % gap from approximate DP vs average run time comparison on the test set of 300 layered graphs. Lower is better for both % gap and run time.

Table 1: Comparison of methods on the synthetic layered graph test set.

| Algorithm | 500- node graphs | | 1000-node graphs | | 2000-node graphs | |
|---|---|---|---|---|---|---|
| | % gap from approx. DP | run time [$s$] | % gap from approx. DP | run time [$s$] | % gap from approx. DP | run time [$s$] |
| Approximated DP | 0 | 264.88 | 0 | 1561.17 | 0 | 8828.86 |
| Depth-First DP (max. run time=1H) | 5.76 | 3600 | 3.84 | 3600 | 2.40 | 3600 |
| Random order | 6.86 | 1.38 | 2.62 | 7.13 | 0.31 | 36.87 |
| Depth-first seq. | 12.9 | 2.45 | 7.1 | 10.91 | 3.57 | 51.32 |
| Breadth-first seq. | 20.94 | 2.43 | 11.31 | 10.87 | 6.42 | 51.52 |
| Neural Topo Order | | | | | | |
| ✓Greedy | 4.32 | 0.6 | 0.48 | 1.19 | -1.47 | 2.44 |
| ✓Sample | 3.49 | 0.72 | **0.03** | 1.41 | **-1.68** | 2.87 |
| ✓Beam search | **3.21** | 2.68 | 0.08 | 5.92 | -1.66 | 14.74 |

## 5.2 Real-World Graphs

While our synthetic layered graphs are convenient for experimentation, we see value in also presenting results obtained from neural computation graphs used for commercial development of our artificial intelligence hardware and software products. Here we sample 115 representative graphs that have diverse architectures (classifiers, language processors, denoisers, etc.) and size (from a few dozen to 1k nodes). We split this dataset into a training set and test set via a random $80 - 20$ split. We train our model for 500 epochs and report the performance on the unseen test set at the end of training in table 2. In order to ensure fair comparison of run times, we stratify the test set into 3 categories based on the graph size. Figure 3 shows the performance vs run time plot on the test set of real graphs. We observe that for real graphs the performance gap between the best baseline (approximate DP) and our model is remarkable. We can obtain sequences which are $50\%$ better than approximate DP on average while also being almost 1000x faster on average. This proves the capability of our model to generalize and perform well on real-world computation workflows.

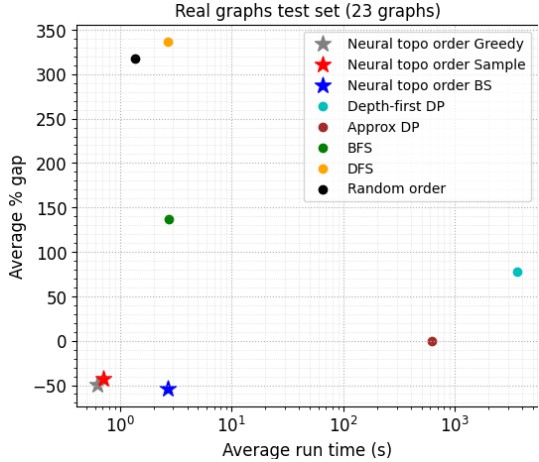

Figure 3: Performance vs run time comparison for different approaches on test set of real computation graphs. Performance is measured in average % gap from approximate DP.

Table 2: Comparison of methods on the real graph test set. Smaller % gap is better

| Algorithm | 200 - 500-node graphs | | 500 - 700-node graphs | | 700 - 1000-node graphs | |
|---|---|---|---|---|---|---|
| | % gap from approx. DP | run time [$s$] | % gap from approx. DP | run time [$s$] | % gap from approx. DP | run time [$s$] |
| Approximated DP | 0 | 113.54 | 0 | 517.60 | 0 | 1131.61 |
| Depth-First DP (max. run time=1H) | 62.18 | 3600 | 102.76 | 3600 | 50.57 | 3600 |
| Random order | 469.34 | 0.25 | 376.16 | 1.24 | 116.24 | 2.40 |
| Depth-first seq. | 506.21 | 0.70 | 394.93 | 2.26 | 123.21 | 4.49 |
| Breadth-first seq. | 348.77 | 0.75 | 149.81 | 2.31 | -35.55 | 4.86 |
| Neural Topo Order | | | | | | |
| ✓Greedy | -17.57 | 0.42 | -51.23 | 0.6 | -68.97 | 0.83 |
| ✓Sample | **-21.53** | 0.44 | -40.51 | 0.68 | -61.46 | 0.97 |
| ✓Beam search | -19.5 | 1.22 | **-57.34** | 2.58 | **-73.45** | 3.86 |

## 5.3 Encoder Ablation Study

We conduct experiments by using an MLP, fully connected transformer and GAT as an encoder architecture to quantify the effectiveness of our topoformer architecture. We test a vanilla version of GAT (referred as GAT forward only) which does message passing only on the edges of the DAG. We also consider GAT encoder which does message passing on the augmented graph having reverse edges corresponding to all the edges of the DAG and refer to this setting as GAT forward+backward.

We train each model on the layered graph dataset of 500 node graphs. We evaluate the performance of the trained model on the test set (300 graphs) of 500 node and 1000 node graphs. We use a sample size and beam width of 16 for evaluation on both 500 and 1000 node graphs. The MLP and transformer use the same number of layers and hidden dimension as the topoformer specified in appendix **??**. We run the inference on our test set of 300 graphs 10 times for each model to be more precise in our run time calculations. We report the mean % gap from approximate DP and the mean run time across all the graphs and trials along with their 95% confidence interval.

Table 3 shows the performance of different encoder architectures. It can be observed that both versions of our topoformer architecture and GAT have a superior performance than MLP and fully connected transformer for both graph sizes. Moreover, full topoformer (message passing on all the seven graphs listed in section 4.1) has a better performance than GAT and topoformer with message

Table 3: Comparison of different encoder architectures. Topoformer with MP (message passing) on DAG corresponds to forward and backward message passing only on the input DAG using topoformer.

| Algorithm | 500-node graphs | | 1000-node graphs | |
|---|---|---|---|---|
| | % gap from approx. DP | run time [$s$] | % gap from approx. DP | run time [$s$] |
| MLP | | | | |
| ✓Greedy | $8.31 \pm 0.76$ | $0.58 \pm 0.0$ | $2.95 \pm 0.48$ | $1.52 \pm 0.01$ |
| ✓Sample | $4.41 \pm 0.50$ | $0.67 \pm 0.0$ | $0.68 \pm 0.35$ | $1.84 \pm 0.02$ |
| ✓Beam search | $6.5 \pm 0.69$ | $2.47 \pm 0.01$ | $2.43 \pm 0.49$ | $7.62 \pm 0.07$ |
| Fully Connected Transformer | | | | |
| ✓Greedy | $8.46 \pm 0.72$ | $0.69 \pm 0.01$ | $3.09 \pm 0.46$ | $1.3 \pm 0.01$ |
| ✓Sample | $4.72 \pm 0.52$ | $0.8 \pm 0.01$ | $0.85 \pm 0.37$ | $1.55 \pm 0.02$ |
| ✓Beam search | $6.52 \pm 0.72$ | $2.98 \pm 0.03$ | $2.09 \pm 0.47$ | $6.49 \pm 0.07$ |
| GAT (forward only) | | | | |
| ✓Greedy | $5.94 \pm 0.61$ | $0.49 \pm 0.01$ | $1.33 \pm 0.38$ | $1.24 \pm 0.01$ |
| ✓Sample | $4.19 \pm 0.56$ | $0.64 \pm 0.01$ | $0.48 \pm 0.36$ | $1.54 \pm 0.02$ |
| ✓Beam search | $4.22 \pm 0.60$ | $2.22 \pm 0.02$ | $0.60 \pm 0.38$ | $5.94 \pm 0.04$ |
| GAT (forward+backward) | | | | |
| ✓Greedy | $4.84 \pm 0.55$ | $0.63 \pm 0.01$ | $0.90 \pm 0.37$ | $1.37 \pm 0.02$ |
| ✓Sample | $3.55 \pm 0.53$ | $0.80 \pm 0.01$ | $0.23 \pm 0.36$ | $1.67 \pm 0.02$ |
| ✓Beam search | $3.55 \pm 0.54$ | $2.89 \pm 0.01$ | $0.39 \pm 0.36$ | $6.50 \pm 0.05$ |
| Topoformer (forward+backward) (Ours) | | | | |
| ✓Greedy | $4.82 \pm 0.55$ | $0.73 \pm 0.01$ | $0.76 \pm 0.36$ | $1.62 \pm 0.02$ |
| ✓Sample | $3.67 \pm 0.52$ | $0.85 \pm 0.01$ | $0.21 \pm 0.36$ | $1.99 \pm 0.02$ |
| ✓Beam search | $3.68 \pm 0.57$ | $3.03 \pm 0.03$ | $0.35 \pm 0.37$ | $8.1 \pm 0.08$ |
| Full Topoformer (Ours) | | | | |
| ✓Greedy | $4.31 \pm 0.56$ | $1.04 \pm 0.01$ | $0.47 \pm 0.36$ | $1.51 \pm 0.01$ |
| ✓Sample | $3.35 \pm 0.52$ | $1.21 \pm 0.01$ | $\textbf{-0.01} \pm 0.35$ | $1.8 \pm 0.02$ |
| ✓Beam search | $\textbf{3.08} \pm 0.51$ | $4.15 \pm 0.02$ | $0.05 \pm 0.36$ | $7.4 \pm 0.07$ |

passing only the forward and backward edges of the DAG. This shows the benefit of global message passing between all the nodes which is enabled by the full topoformer.

## 6 Conclusion

In this work we propose an end-to-end machine learning method for the task of optimizing topological orders in a directed acyclic graph. Two key elements in our design are: (1) an attention-based GNN architecture named Topoformer that employs message passing that is both global and topologically-aware in directed acyclic graphs, (2) a non-autoregressive parametrization of the distribution on topological orders that enables fast inference. We demonstrated, for both synthetic and real-world graphs, the effectiveness of the method in tackling the problem of minimizing peak local memory usage for a compute graph – a canonical task in compiler pipelines. Said pipelines also include other tasks [12], chief amongst them the one of assigning operations to devices for execution. At the present stage, our method and dataset cannot be leveraged for solving these, or for end-to-end optimization of a whole pipeline. Extending our method to this more challenging setting is therefore a natural direction for future research.

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
