# Appendix

## A   Layered graphs dataset

We report here the details of the generation algorithm we use to create our dataset. It is not the first time that a synthetic dataset of graphs is used to train and test an ML framework on a compiler task, as this was already done in ref. [11]. However, the models therein used were generic random graph models (e.g. Erdos-Renyi), rather than a model explicitly tailored to reproduce NN-like computation graphs. We develop such a model, and we release its details with the intent of both ensuring reproducibility of our results, as well as of providing tool that we hope will be picked up by researchers interested in compiler problems, as well as more general sequence optimization task on DAGs.

The algorithm builds a graph by organizing a fixed number $|V|$ of nodes into well-defined layers, and then placing edges between subsequent layers, as well as skip connections that skip at least one layer. While the number of nodes is fixed by the user, the target number of layers $L$ depends on the *width factor* $\mathcal{W}$ of the graph. A width factor of 0 would result in a one-dimensional chain graph, whilst a width factor of 1 in a graph with a single, wide layer,

$$L = \left\lceil \sqrt{|V|\left(\frac{1}{\mathcal{W}} - 1\right)} \right\rceil, \tag{11}$$

where $\lceil \cdot \rceil$ is the ceiling function. In order to promote architectural variability within the dataset, we choose to randomly draw a new width factor, $\mathcal{W} \sim U(0.25, 0.5)$, for each graph, with $U(a, b)$ denoting the uniform distribution in the $[a, b]$ interval. Subsequently, the number of nodes to assign to each layer $\ell$ is also an integer randomly drawn from a uniform distribution

$$\mathcal{N}_\ell \sim U\left(\lceil |V|/L\left(1 - \sigma_\mathcal{N}\right)\rceil, \lfloor |V|/L\left(1 + \sigma_\mathcal{N}\right)\rfloor\right), \tag{12}$$

with $\sigma_\mathcal{N}$ being a user-defined variability parameter, and $\lfloor \cdot \rfloor$ is the floor function. We stress that both $L$ and $\mathcal{N}_\ell$ are just target values, since we wish to keep $|V|$ fixed: this layer-by-layer node addition process is stopped as soon as the graph has the number of nodes $|V|$ required, which might lead to the number of layers and nodes per layer being ultimately different from their respective targets. The pseudocode for this procedure is reported in algorithm 1.

---

**Algorithm 1:** Node-assignment algorithm for layered graphs.

---

**Output:** A layered graph $G = (V, )$ without edges
**Input:** Total number of nodes $|V|$, number-of-nodes-per-layer variability $\sigma_\mathcal{N}$
**Data:** layer index $\ell$, node index $n$, node counter $N$, target number $\mathcal{N}_\ell$ of nodes for layer $\ell$
$\ell \leftarrow 0$;
$N \leftarrow 0$;
**while** *True* **do**
    $\mathcal{N}_\ell \sim U\left(\lceil |V|/L\left(1 - \sigma_\mathcal{N}\right)\rceil, \lfloor |V|/L\left(1 + \sigma_\mathcal{N}\rfloor\right)\right)$;
    **for** $n \in [1, \mathcal{N}_\ell]$ **do**
        **if** $N \geq |V|$ **then**
            break;
        add node $n$ to graph $G$;
        add node $n$ to layer $\ell$;
        $N \leftarrow N + 1$;
    **end**
    $\ell \leftarrow \ell + 1$
**end**

---

After the layers are set up, the algorithm proceeds to assign edges between adjacent layers. As an example, let us assume that $\mathcal{N}_1$ and $\mathcal{N}_2$ are the numbers of nodes for two adjacent layers, with

$\mathcal{N}_2 < \mathcal{N}_1$. The maximal number of edges between these two layers, corresponding to a fully-connected, MLP-like topology, would be $\mathcal{N}_1 \times \mathcal{N}_2$. Since we want each node to have at least one ingoing and one outgoing connection (except for those in the first and last layers), the minimal number of connections must be $\max(\mathcal{N}_1, \mathcal{N}_2) = \mathcal{N}_1$. The user can interpolate between these two extrema by tuning the *edge density* parameter $\rho_E$, with the number of edges to place between the two layers being ultimately equal to

$$|E|_{(\ell_i, \ell_{i+1})} = (\mathcal{N}_{\ell_i} \times \mathcal{N}_{\ell_{i+1}})\rho_E + (1 - \rho_E)\max(\mathcal{N}_{\ell_i}, \mathcal{N}_{\ell_{i+1}}). \tag{13}$$

This budget of edges is subsequently distributed among the nodes in the larger layer (layer 1 in our example), with them being assigned to the node with the smallest number of so-far-assigned edges (ties are broken randomly), until it is exhausted. What then remains to do is connecting all the so assigned edges to nodes in the other layer (layer 2 in our example above). We choose these destination nodes in a such a way that, if the layers were visualized as being centered one above the other, with the larger layer at the top, the edges assigned to a node end up more or less equally spaced in 2-$d$ cone below it. This procedure is repeated for every pair of adjacent layers, as we report in algorithm 2.

---

**Algorithm 2:** Edge-assignment algorithm for layered graphs.

---

**Output:** A layered graph $G = (V, E)$ with edges but no skip connections.
**Input:** A layered graph $G = (V, )$ without edges, edge density $\rho_E$
**Data:** Number $|E|_{(\ell_i, \ell_j)}$ of edges between layers $\ell_i$ and $\ell_j$. $c_n$ is a counter of edges incoming or outgoing from node $n$

**for** $\ell_1 \in$ *graph layers* **do**

    $\ell_2 = \ell_1 + 1$;

    $|E|_{(\ell_1, \ell_2)} = (\mathcal{N}_{\ell_1} \times \mathcal{N}_{\ell_2})\rho_E + (1 - \rho_E)\max(\mathcal{N}_{\ell_1}, \mathcal{N}_{\ell_2})$ (rounded to the closest integer);

    **if** $\mathcal{N}_1 \geq \mathcal{N}_2$ **then**

        $\ell_s \leftarrow \ell_1, \ell_t \leftarrow \ell_2$;

    **else**

        $\ell_s \leftarrow \ell_2, \ell_t \leftarrow \ell_1$;

    **end**

    **for** $n \in \ell_s$ **do**

        $c_n \leftarrow 0$;

    **end**

    **while** $\sum_{n \in \ell_s} c_n < |E|_{(\ell_1, \ell_2)}$ **do**

        $\mathcal{S} \leftarrow \arg\min c_n$;

        Pick $i$ randomly from set $\mathcal{S}$;

        $c_i \leftarrow c_i + 1$;

    **end**

    **for** $n \in [0, \mathcal{N}_{\ell_s} - 1]$ **do**

        **if** $\mathcal{N}_{\ell_s} = 1$ **then**

            $n_c = 0$;

        **else**

            $n_c = n \times \frac{\mathcal{N}_{\ell_t} - 1}{\mathcal{N}_{\ell_s} - 1}$ set "center node," rounded to the nearest integer

        **end**

        **for** $i \in [0, c_n - 1]$ **do**

            $n_t = (n_c - (c_n - 1)//2) + [0, c_n - 1]$ (a range centered at $n_c$);

            Shift the range $n_t$ up/down such that no index is less than 0 or greater than $\mathcal{N}_{\ell_t} - 1$ ;

            **for** $j \in n_t$ **do**

                add one edge between node $n$ of layer $\ell_s$ and node $j$ of layer $\ell_t$

            **end**

        **end**

    **end**

**end**

---

Skip connections, i.e. edges skipping at least one layer, which are often found in modern NN architectures, are then added to the graph. The total number of skip connections to add is fixed as

$$\mathcal{N}_S = |E| \frac{\rho_S}{(1 - \rho_S)}, \tag{14}$$

where $|E|$ is the total number of edges in the graph so far, and $\rho_S$ a user-defined skip connection density. For each skip connection, we randomly draw a source layer among those between the first and the third-to-final ones (since skip connections must skip at least one layer). The target layer number is then also drawn at random between the source layer number $+2$, and the final layer (both included). One must then assign a source and a target *node* within each of these layers. We just select the source node at random within the source layer, and then assign the target node in such a way that it would be more or less directly below the source node if the graph were visualized on a 2-$d$ plane. The pseudocode of this procedure is reported in algorithm 3, and figure 4 shows three example instances of layered graphs created with our algorithm.

---

**Algorithm 3:** Skip connection-assignment algorithm for layered graphs.

---

**Output:** A layered graph $G = (V, E)$ with both connections between adjacent layers, and skip
$\qquad$ connections
**Input:** A layered graph $G = (V, E)$ with edges between adjacent layers but no skip connections,
$\qquad$ skip connection density $\rho_S$
**Data:** number of layers $L$, number of edges $|E|$
**if** *L<3* **then**
$\qquad$ break; /* cannot have skip connections with fewer than 3 layers $\qquad$ */

$\mathcal{N}_S = \lceil |E| \frac{\rho_S}{(1-\rho_S)} \rceil$;
**for** $i \in [0, \mathcal{N}_S)$ **do**
$\qquad \ell_s \leftarrow$ a layer at random between the first and third-to-last (both included);
$\qquad \ell_t \leftarrow$ a layer at random between layer number $\ell_s + 2$ and the last (both included);
$\qquad x_s \sim U(0, 1)$;
$\qquad y \sim U(0, 1)$;
$\qquad x_t = x_s + 0.2 \times y$;
$\qquad x_t = \min(x_t, 0.999)$; /* ensure that $x_t \in [0, 1)$ $\qquad$ */
$\qquad$ add an edge between node $\lfloor x_s \mathcal{N}_{\ell_s} \rfloor$ of layer $\ell_s$ and node $\lfloor x_t \mathcal{N}_{\ell_t} \rfloor$ of layer $\ell_t$
**end**

---

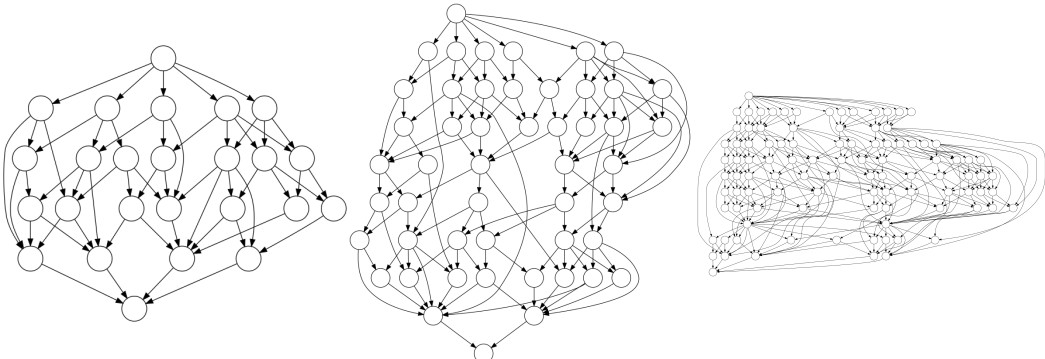

Figure 4: Three example graphs from the layered graph family with (from left) 25, 50, and 100 nodes, generated using the algorithm we describe in the text. One can clearly make out the layered structure and easily remark the presence of skip connections.

Finally, we specify the assignment of memory costs to the nodes. In the layered graph model, we have both output memory costs $(m_i)_{i=1}^{|V|}$ and parameter costs $(p_i)_{i=1}^{|V|}$, where the output cost is the memory usage of the output of an operation, and the parameter cost the one of a variable necessary to execute the operation; for example, if the operation at node $i$ were a matrix multiplication, $\boldsymbol{y} = M\boldsymbol{x}$, $o_i$ would be the memory usage of $\boldsymbol{y}$ and $p_i$ the one of the matrix $M$. The parameter cost of operation $\sigma_t$ during a sequence is added to the memory usage at time $t$, but not to the cost at subsequent steps since the memory associated to it can be de-allocated as soon as the operation has been executed.In particular, the memory utilization cost $M_t$ in (2) gets modified to the following:

$$M_t = I_{t-1} + m(\sigma_t) + p(\sigma_t) \tag{15}$$

where $I_t$ is defined in (3). Both costs are randomly drawn from a simple mixture of Gaussians $\text{GMM}(\mathbf{w}, \mu, \sigma) \equiv \sum_{i=1}^{4} w_i \mathcal{N}(\mu_i, \sigma_i)$ projected on to the positive reals,

$$m_i \sim \text{GMM}(\mathbf{w}, \mu, \sigma)\big|_{\mathbb{R}_+}, \quad p_i \sim \text{GMM}(\mathbf{w}, \mu, \sigma))\big|_{\mathbb{R}_+}. \tag{16}$$

To align the costs assignment with the real world computation graphs, instead of sampling the memory costs for each node $n$, we sample one output cost $m_l$ and parameter cost $p_l$ for each layer $l$ and assign the costs $m_l, p_l$ to each node in layer $l$. This is because many real world computation graphs are a tiled version of the original precedence graph of compute nodes where each node is broken down into a layer of nodes with similar shape and parameter requirements. This concludes the description of our dataset generation algorithm. For the sake of reproducibility, we report below the value we took for all the user-defined parameters mentioned in this section:

- Variability of number of nodes per layer $\sigma_{\mathcal{N}} = 0.75$
- Edge density $\rho_E = 0.2$
- Skip connection density $\rho_S = 0.14$
- Means of the Gaussian mixture $(\mu_1, \mu_2, \mu_3, \mu_4) = (0.5, 1, 3, 5)$
- Standard deviations of the Gaussian mixture $(\sigma_1, \sigma_2, \sigma_3, \sigma_4) = (0.5, 1, 1, 1)$
- Weights of the Gaussian mixture $(w_1, w_2, w_3, w_4) = (0.3, 0.3, 0.3, 0.1)$

## B    Decoder Ablation study

In order to measure the effectiveness of our architecture we perform ablation experiments to study the effect of changing the decoder to an auto-regressive decoder and changing both the encoder and the decoder (Table 4).

We compare the performance of our architecture with the model which uses topoformer as an encoder but uses an auto-regressive decoder. We adapt the decoder designed for the TSP problem [6] for our memory-minimization problem. The decoder of [6] uses a notion of context node for decoding and at each decoding step using a series of multi-head attention with the context node arrives at the distribution of the next node to be selected for the order. We modify the masking procedure in the decoder of [6] to mask out all the nodes which are not present in the set of feasible next nodes $\mathcal{S}(\sigma_{1:t-1}, G)$.

We also conduct an experiment by changing both the encoder and decoder by adapting the model of [6] to our problem. We adapt the auto-regressive decoder of [6] as described above. [6] uses a fully connected transformer as an encoder since the underlying graph in TSP is a fully connected graph. We modify the encoder of [6] to do message passing only on the edges of our input DAG so that it can exploit the topological structure of the graph in the encoding stage. We refer to this model as "GNN encoder + AR decoder" in table 4.

We train both the models: "GNN encoder + AR decoder" and "Topoformer + AR decoder" on the layered graph dataset of 500 node graphs. We evaluate the performance of the trained model on the test set (300 graphs) of 500 node and 1000 node graphs. We use a sample size and beam width of 16 for 500 node graphs and a sample size and beam width of 8 for 1000 node graphs. We use a smaller sample size for 1000 node graphs due to GPU memory issues with the auto-regressive decoder approaches.

Table 4 shows the mean and the 95% confidence interval of the % gap from approximate DP and run time for the three approaches on 500 and 1000 node graphs. We note that the performance of topoformer with AR decoder is quite close to our model for both 500 and 1000 node graphs. However, our model can run inference 2x faster than topoformer with AR decoder on 1000 graphs nodes (in greedy mode). Also, our model outperforms the adaptation of [6] attention based GNN encoder and AR decoder to our problem both in terms of memory cost of sequence and run time. This shows the merit of our topoformer architecture over using a traditional GNN architecture which does message passing only on the input graph.

Table 4: Comparison with Auto-regressive decoding

| Algorithm | 500-node graphs | | 1000-node graphs | |
|---|---|---|---|---|
| | % gap from approx. DP | run time [s] | % gap from approx. DP | run time [s] |
| GNN encoder + AR decoder | | | | |
| ✓Greedy | $6.13 \pm 0.58$ | $1.66 \pm 0.01$ | $1.84 \pm 0.39$ | $3.34 \pm 0.02$ |
| ✓Sample | $4.71 \pm 0.56$ | $1.76 \pm 0.01$ | $1.38 \pm 0.37$ | $3.59 \pm 0.02$ |
| ✓Beam search | $4.87 \pm 0.61$ | $4.01 \pm 0.02$ | $2.09 \pm 0.41$ | $7.90 \pm 0.05$ |
| Topoformer + AR decoder | | | | |
| ✓Greedy | $4.43 \pm 0.55$ | $1.53 \pm 0.01$ | $0.53 \pm 0.35$ | $3.05 \pm 0.02$ |
| ✓Sample | $3.33 \pm 0.51$ | $1.7 \pm 0.01$ | $\mathbf{0.05} \pm 0.35$ | $3.38 \pm 0.02$ |
| ✓Beam search | $3.14 \pm 0.52$ | $4.27 \pm 0.04$ | $0.13 \pm 0.36$ | $7.90 \pm 0.05$ |
| Topoformer + NAR decoder (Ours) | | | | |
| ✓Greedy | $4.31 \pm 0.56$ | $1.04 \pm 0.01$ | $0.47 \pm 0.36$ | $1.53 \pm 0.01$ |
| ✓Sample | $3.35 \pm 0.52$ | $1.21 \pm 0.01$ | $0.09 \pm 0.35$ | $1.78 \pm 0.01$ |
| ✓Beam search | $\mathbf{3.08} \pm 0.51$ | $4.15 \pm 0.02$ | $0.2 \pm 0.36$ | $5.57 \pm 0.05$ |

## C   Training and Model details

### C.1   Training

We train our model using the ADAM optimizer with the initial learning rate of $10^{-4}$ and learning rate decay factor of $0.996$ per epoch. We use a batch size of $8$ for training our model. The training and testing of our model is done on a single GPU (Nvidia Tesla V-100) with 32 GB memory. We trained our model for 326 epochs on the synthetic graph dataset where in each epoch we provide 1000 training graphs. Also, we provide a new training set in each epoch so that we do not overfit our model on a fixed training set. We found the training to be fairly stable, and it converged in about 1-2 days.

### C.2   Model architecture

We use topoformer with number of layers $n_{layers} = 4$, embedding dimension $d = 256$, number of heads $n_{heads} = 10$ for each MHA operation on the seven graphs listed in section 4.1 and the query and value dimension of $64$ for each head of MHA. The MLP used in (5) consists of a linear layer ($d_{input} = d_{output} = 256$) with GELU activation followed by another linear layer ($d_{input} = d_{output} = 256$). The MLP used in (7) to generate the node priorities consists of a linear layer ($d_{input} = d_{output} = 256$) with RELU activation followed by another linear layer ($d_{input} = 256, d_{output} = 1$). In order to restrict the range of priority values, we also normalize the priorities of the nodes used for the decoding as follows:

$$\tilde{y}_i = \alpha \times \frac{y_i - \text{mean}(\mathbf{y})}{\text{std}(\mathbf{y})} \tag{17}$$

where $\mathbf{y} = \left[ y_1, y_2, \ldots, y_{|V|} \right]$ and $\alpha$ is a hyperparamter. We set $\alpha = 5$ for our experiments.

### C.3   Baselines

We provide more details about the dynamic programming baselines used in our experiments to compare the performance of our model

- **Depth-First Dynamic Programming (DP).** Topological orders are generated in a depth-first manner (with backtracking) where next node is picked randomly among available candidates. Branch exploration is terminated if 1) the same set of nodes are in the partial sequence as a branch that has been already explored - only the lowest cost partial sequence is retained (dynamic programming approach), and 2) if the current partial cost is already higher than the lowest cost of any full sequence already found (cost increases monotonically). This algorithm will eventually find the global optimal order, though the run time for doing so is

expected to be at least exponential in |V| [2]; it is however able to return at least one complete sequence in time $O(|V| + |E|)$ [34] in the worst case, same as DFS. In our implementation, we set a wall time of one hour and pick the best complete path found. We observe that for our synthetic layered graphs, if the graph size is as small as $|V| = 100$, we can actually find the optimal sequence in most cases within the one hour budget. We ran this algorithm on a CPU machine with Intel(R) Xeon(R) W-2123 CPU @ 3.60GHz

- **Approximate DP.** We define the state space $S$ as the space including a set of all nodes for each partial sequence (which *ignores* the ordering information) and the action space for each state as the space of all possible next-node choices at that state (based on the topological structure). As an example for the state representation, if there is a partial sequence $5 \to 2 \to 4 \to 3 \to 1$, the corresponding state is $\{1, 2, 3, 4, 5\}$. With the empty set $\emptyset$ being an initial state (meaning that no node has been added), we consider a state transition model that adds an action (a node) to a state and creates a successor state. Specifically, we can partition $S$ into $S_0 \cup S_1 \cup \cdots \cup S_{|V|}$, where $S_t$ is the space including a set of all nodes for each length-$t$ partial sequence (note that $S_0 = \{\emptyset\}$). At every iteration $t = 0, 1, ..., |V| - 1$, the algorithm takes $S_t$ and assumes that we have *(1) the minimum cost* and *(2) the best partial sequence* for each state in $S_t$, where the minimum cost is over all feasible partial sequences corresponding to the state. Then, for each successor state in $S_{t+1}$, the algorithm computes the minimum cost and the best partial sequence for reaching out that state.
  It should be noted that the algorithm gives an *exact* solution if the amount of time and memory resource is sufficient, e.g., an exact solution can be found for 100-node graphs. However, due to the practical resource limitation, we only keep top-$K$ elements of $S_{t+1}$ for each iteration $t$ based on costs. We use the beam size $K = 100,000$ for all experiments, and Nvidia Tesla V-100 is used for parallel computation across multiple states for each iteration.

## C.4   Baseline policy

The baseline $b(G)$ used in the policy gradient update is generated using the greedy rollout of the baseline policy. The baseline policy is also an instance of our model which is updated regularly during the course of training. At the end of each epoch, if the performance of the model being trained becomes better than the baseline model (in greedy inference mode) on a set of validation graphs then we copy the weights of the trained mode to the baseline model.

## C.5   Input features and initial node embedding

We use the following as the input features $x_j$ for node $j$:

1. Output memory cost $m_j$ and parameter memory cost $p_j$
2. In-degree and out-degree of the node
3. Minimum and maximum distance (in terms of hop count) of the node from the source and target node

We normalize each entry of the input node feature across the nodes so that the features lie between $0$ and $1$ making it invariant with respect to the graph size. To be precise, the $i^{th}$ entry of the normalized input feature of node $j$ is given as $\bar{x}_j^i = \frac{x_j^i}{\max_n x_n^i}$. We also augment the node features with the Laplacian positional encodings (PE) [35] of dimension 20. We compute the laplacian PE using the laplacian matrix of the undirected DAG where all the directed edges are converted to undirected edges. Finally, the initial embedding $h_j^0$ for node $j$ is obtained by passing $\bar{x}_j$ through a linear layer.