# OpenReview forum: "Neural Topological Ordering for Computation Graphs"
_NeurIPS.cc/2022/Conference — NeurIPS 2022 Accept_

### Official Review · Reviewer_BLAK · 2022-06-20

**Rating:** 6
**Confidence:** 5
**Soundness:** 4 excellent
**Presentation:** 4 excellent
**Contribution:** 3 good

**Summary:**

This paper addresses topological ordering over computational graphs, and proposes a neural model, the Topoformer, to compute an optimal topological ordering (in terms of peak memory usage) when compiling a directed acyclic graph of compute operations, e.g., in neural networks. At a high level, the Topoformer builds on graph neural network (GNN) architectures, and applies these models simultaneously to multiple graph structures (e.g., the transitive reduction of the input DAG) using separate multi-head attention mechanisms, and then combines the attention outputs using MLPs. The Topoformer uses a simple (albeit less powerful) decoder, based on computing node priorities using an MLP layer on the final node embeddings, rather than an auto-regressive approach, so as to efficiently compute node sequences in one model pass. Finally, the paper evaluates the Topoformer (with sampling, greedy and beam search decoding), and against multiple baselines (random orderings, dynamic programming, etc.) on both synthetically generated neural network baselines, as well as real-world baselines. In all experiments, the Topoformer model achieves comparable performance to the best baseline (Approximate DP), and eventually outperforms it on larger instances, even with a fraction of the computational resources.

**Questions:**

No questions at present.

**Limitations:**

Limitations: See final bullet in "Weaknesses" section above.

Societal Impact: Not applicable

**Strengths And Weaknesses:**

Strengths:
- The paper is well-written, and the results are compelling.
- The model is highly scalable and easy to understand. Furthermore, its fundamental idea of using multiple graph representation is an interesting direction for graph data augmentation in general.
- The dataset generation procedure is potentially valuable for the MLCO community, and can serve as a strong baseline aside from the more traditional baselines such as TSP.

Weaknesses:
- The comparison against baselines is rather limited: It would be good to validate the strengths of this approach against different GNN choices, such as DAGNN, or even standard GNNs such as GCN, GAT. Currently, the baselines are all non-neural. It would therefore be insightful to include other GNN results to demonstrate the value of the Topoformer architecture.
- The analysis of the model could be improved: For instance, it would be nice to see how the different attention components combine, e.g., a case study on a representative test instance, or to conduct an ablation study of the different graph structures included as input to the Topoformer.
- The limitations of the one-shot decoder are not sufficiently discussed. The paper states that expressiveness is sacrificed, but does not thoroughly discuss the potential drawbacks, namely loss of potential ordering outputs. Ideally, one should highlight scenarios in which this choice could be harmful, e.g., some DAG where the optimal sequence cannot be obtained, or, alternatively, whether this limitation is not impactful in practice.

---

> ### Author Response · Authors · 2022-08-02
> **Response to Reviewer BLAK**
>
> We thank Reviewer BLAK for the constructive feedback. We try to address your concerns below:
>
> 1. **The comparison against baselines is rather limited: It would be good to validate the strengths of this approach against different GNN choices, such as DAGNN, or even standard GNNs such as GCN, GAT. Currently, the baselines are all non-neural. It would therefore be insightful to include other GNN results to demonstrate the value of the Topoformer architecture.**
>
>     In our original submission, we compared our approach to neural baselines in the Supplementary Material but didn't explicitly refer to it in the main paper. We made it more explicit in the revision. We study the effect of changing the encoder/decoder architecture while keeping one of them fixed, as well as one ablation study on effect of changing both encoder and decoder simultaneously. Based on reviewer's feedback, we have also added an additional ablation study which employs GAT as an encoder to the revised manuscript. Please see the general response on ablation studies for more details.
>
> 2. **The analysis of the model could be improved: For instance, it would be nice to see how the different attention components combine, e.g., a case study on a representative test instance, or to conduct an ablation study of the different graph structures included as input to the Topoformer.**
>
>     We did an ablation study of Topoformer (forward+backward) as in the aforementioned response to all the reviewers and find that although it is still better than the other encoder architectures, there is some performance drop compared to using all the constituent graphs in full Topoformer. This shows the benefit of global message passing between all the nodes which is enabled by the full Topoformer.
>     We agree with the reviewer that considering more granular ablations would be useful, but given the time and resource constraints we were not able to carry these out.
>
> 3. **The limitations of the one-shot decoder are not sufficiently discussed. The paper states that expressiveness is sacrificed, but does not thoroughly discuss the potential drawbacks, namely loss of potential ordering outputs. Ideally, one should highlight scenarios in which this choice could be harmful, e.g., some DAG where the optimal sequence cannot be obtained, or, alternatively, whether this limitation is not impactful in practice.**
>
>     We wish to clarify our statement concerning the loss in expressive power; we do not state outright that expressive power is lost, but rather that such a loss would be acceptable given the benefit in runtime that NAR/one shot decoding affords. And even then, by "expressivity'' we only refer to the capability of our decoding scheme to approximate any probability distribution over sequences, and not necessarily to its capability to parametrize a probability distribution whose mode corresponds to the optimal sequence, which actually determines performance.
>
>    Our experiments indeed shows that if our decoding scheme leads to any loss in expressivity, this has anyway a negligible impact on performance. In the Supplementary Material we report an experiment wherein decoding is carried out via the AR decoder of [1] while leaving the encoder unchanged; we observe a negligible drop in performance, while speeding up runtime by almost 2x with respect to the AR scheme.
>     In summary, while we consider it acceptable to trade pure expressivity for runtime, we do not expect, and indeed we do not observe, a significant performance/runtime tradeoff.
>
>
> References:
>
> [1] [W. Kool, H. van Hoof, and M. Welling, “Attention, learn to solve routing
> problems!” in *International Conference on Learning Representations, 2019*](https://arxiv.org/abs/1803.08475)

---

> > ### Comment · Reviewer_BLAK · 2022-08-05
> > **Reviewer Response**
> >
> > Thank you for your response.
> >
> > 1. Thanks for the added experiments. This makes things clearer. It's also good that these are now referenced in the main body.
> > 2.  My point here is more conceptual, rather than empirical. Here I was looking for some theoretical arguments justifying why the Topoformer is performing better or showing an example highlighting the limitations discussed. Right now, this point is left a bit too vague in my opinion.
> > 3. >  we do not state outright that expressive power is lost, but rather that such a loss would be acceptable given the benefit in runtime that NAR/one shot decoding affords. And even then, by "expressivity'' we only refer to the capability of our decoding scheme to approximate any probability distribution over sequences, and not necessarily to its capability to parametrize a probability distribution whose mode corresponds to the optimal sequence, which actually determines performance.
> >
> > I do not understand your point here, I'm afraid. Surely, if your model cannot fit any arbitrary probability distribution, then there exists a distribution which it cannot represent well, and then providing this as input to your model would not yield good results. Hence, one cannot separate these two points: It could well be that the optimal sequence corresponds to some distribution beyond the expressive power of your model. Of course, I understand how the experiments show the empirical strength of the model and don't indicate any performance loss, but my comment here relates to a deeper understanding of these distributions that you say cannot all be captured. As a result, this point remains unclear to me.
> >
> > At this moment, I will keep my current rating.

---

> > > ### Author Response · Authors · 2022-08-09
> > > **Response to Reviewer BLAK**
> > >
> > > We appreciate your valuable feedback, and we are willing to clarify our statement regarding point number 3. As we wish to stress, given how the policy is used at inference time, there is no one-to-one correspondence between the optimal sequence and a distribution; there is only a correspondence between the optimal sequence and the mode (i.e. the most likely value of the random variable, which is related to our greedy inference) of the distribution. Multiple distributions can have the same mode, but all these distributions will be equivalent in terms of final performance, given the way in which that we quantify it.
> > > Within this set of distributions which share the same mode, some will likely be impossible for our NAR decoding-based model to approximate; but as long as some others in the set can be approximated, we do not expect a performance drop.
> > >
> > > We make this more concrete by the following analysis. For any given graph $G$ and the optimal sequence $s^*(G)$, one can easily show that there always exists a node priority that yields the optimal sequence. For example, let $I_j^*(G)$ be the node $j$'s index in the sequence $s^*(G)$. If we let $y_j(G)=|G|-I_j^*(G)$ be the node priority of node $j$ and generate a sequence by following that greedily, the resultant sequence will be $s^*(G)$. Actually, one can find out infinitely many priority values which can lead to $s^*(G)$; as long as the node priority decrease when $I_j^*(G)$ increase, the optimal sequence $s^*(G)$ can be obtained.
> > >
> > > Suppose we train our model $p(y|G)$ on a single graph $G$, where $y$ is the node priority as described in our paper. Based on the aforementioned one-to-infinitely-many relation between the optimal sequence and optimal node priorities, training the model to output any one of those node priorities will lead to the optimal sequence via greedy inference. Also, since our model $p(y|G)$ with neural networks can generate arbitrary node priorities, we can conclude that our model is sufficiently powerful to find out the optimal sequence in a single graph task.
> > >
> > > Now, let us discuss training over an ensemble of graphs. Since a graph $G$ is provided as a conditioner to compute the priority values $y$ (from $p(y|G)$), we can generate a set of optimal priority values which correspond to the optimal sequence for each graph via greedy inference.
> > >
> > > Hence there is no performance loss when using our proposed decoding method as long as the encoder has enough expressive power.

---

### Official Review · Reviewer_hyRi · 2022-07-11

**Rating:** 7
**Confidence:** 2
**Soundness:** 3 good
**Presentation:** 3 good
**Contribution:** 3 good

**Summary:**

The authors introduce an architecture called Topoformer. It employs message passing that is both global and topologically- aware in directed acyclic graphs. It is an end-to-end framework of encoder and decoder for topological ordering focusing on the compiler task of optimizing the peak local memory usage during execution. Experiments are conducted on both real and synthetic graphs of varied sizes.

**Questions:**

1. What about graphs with more than 2k nodes?
2. Can we show comparison wrt recent algorithms for the same problem?

**Limitations:**

Yes

**Strengths And Weaknesses:**

Strengths:
1. The problem is quite novel and can be a notable contribution to the field of GNNs, CO problems and compilers.
2. Experiments show improved results.
3. Paper is mostly explained well, sometimes difficult to follow.
4. Some of the results are interesting.
5. Problem is well placed in the literature and impactful.
6. To the best of my knowledge, this is the first work based on GNN+topological ordering in DAG.

Weaknesses:
1. Paper is sometimes difficult to follow.
2. Example figures should have been included.
3. Some more analyses and explanations could be performed to show the significance of the attention mechanism or Transformer network.
4. I don’t see any comparison with SOTA.

---

> ### Author Response · Authors · 2022-08-02
> **Response to Reviewer hyRi**
>
> We thank Reviewer hyRi for the constructive feedback. We try to address your concerns below:
>
> 1. **Paper is sometimes difficult to follow.**
>
>       We would like to improve our manuscript and would appreciate pointers to sections where clarity could be improved.
>
> 2. **Example figures should have been included.**
>
>     We have example visualizations of synthetic graphs in Supplementary Material.
>
> 3. **Some more analyses and explanations could be performed to show the significance of the attention mechanism or Transformer network.**
>
>     Please see the general response for the ablation study that we carried out.
>
> 4.  **I don’t see any comparison with SOTA.**
>
>       There is no SOTA algorithm for the task we consider here; while there are relevant papers in this field [1,2] neither of them makes available the code or the dataset to reproduce their respective results on memory usage optimization; a synthetic dataset is made available in [2], but it is not used for the task we consider here. We hope that our detailed description of the synthetic graph dataset will provide an industry wide benchmark dataset for this problem.
>
> 5. **Can we show comparison wrt recent algorithms for the same problem?**
>
>       Please see answer to question 4.
>
> 6. **What about graphs with more than 2k nodes?**
>
>       We can run our model on graphs with more than 2k nodes, but as the graph size increases the memory requirement of our model also increases. Also, for larger graphs one should also increase the time budget of the DP baselines, as the number of topological orders increase exponentially with graph size. Keeping these constraints in mind, we restricted our experiments with graphs with up to 2k nodes.
>
>
> References:
>
> [1]  [Y. Zhou, S. Roy, A. Abdolrashidi, D. Wong, P. Ma, Q. Xu, H. Liu,
> P. Phothilimtha, S. Wang, A. Goldie et al., “Transferable graph optimiz-
> ers for ML compilers,” *Advances in Neural Information Processing Systems,
> vol. 33, pp. 13 844–13 855, 2020*](https://proceedings.neurips.cc/paper/2020/hash/9f29450d2eb58feb555078bdefe28aa5-Abstract.html)
>
> [2]  [A. Paliwal, F. Gimeno, V. Nair, Y. Li, M. Lubin, P. Kohli, and O. Vinyals,
> “Reinforced genetic algorithm learning for optimizing computation graphs,”
> in *International Conference on Learning Representations, 2020.*](https://arxiv.org/abs/1905.02494)

---

### Official Review · Reviewer_bMHc · 2022-07-19

**Rating:** 6
**Confidence:** 4
**Soundness:** 3 good
**Presentation:** 4 excellent
**Contribution:** 3 good

**Summary:**

The authors introduce a new approach for solving the problem of topological orderings on DAGs with additional ordering constraints. They do so using a novel neural architecture based on a GNN with attention. The authors focus on peak memory minimization as a setting for application — specifically, computation graphs generated by neural network computation. They present a simple but effective method of generating layered graphs which serve as synthetic data for their benchmarks, pointing out several challenges with real-world benchmarks, then benchmarking their approach on both, which shows significant improvements in runtime over other approximate methods.

**NB:** I am reviewing this paper as an emergency substitute; please let me know if you have questions about my review or anything I can clarify.

**Questions:**

The decision to add constituent graphs as other attention heads is sensible, but there is no speculation or study as to why that inductive bias is effective. What other approaches did the authors explore, and why were they successful or not?

Is it possible this approach could be used to solve other problems in P# or related complexity classes that have reductions to or from this particular constrained topological ordering problem? Additional discussion of this might be compelling.

Beam search significantly increases the cost of inference for the model; results, however, don’t seem much better. Are there any insights as to why beam search doesn’t yield much improvement?

Have the authors verified that the orderings produced by the model work in practice? A real experiment using the constructed computation graphs would be very compelling.

Per the above: what is the model size, and how long does it take to train on what scale of hardware? Is training sensitive to ay configurations or parameters?

**Strengths And Weaknesses:**

### Strengths

The proposed method is novel. In particular, the introduction of additional attention heads of component graphs of the fully-connected graph is a unique way to inject inductive bias into the model.

Many performance considerations are made, then challenges mitigated. Using a well-crafted non-autoregressive approach significantly speeds up inference for large graphs.

The paper is very well-written and pleasant to read. Its premises are compellingly argued. The motivation and problem setting are clear and the utility simple.

### Weaknesses

The paper lacks ablations on some of the architectural contributions – some decisions seem somewhat arbitrary as a result. As an example: why is an additional node-wise MLP layer added to follow MHA sub-layers in the model? Is this layer a prerequisite to training, or is it simply a change which empirically improved performance?

Details on the generation of the real-world graphs are lacking. Further, the authors do not discuss releasing the set of graphs you used to evaluate your model, which would greatly enhance reproducibility and would further the paper’s contribution.

The discussion of the approaches for generating and the origins of the real-world graphs can be significantly improved to better-understand the differences between the real-world and generated graphs.

The authors can do a better job of broadening the treatment of the benefits of the proposed approach to other types of problems in topological ordering. The problem setup and study, while useful, is rather narrow.

Finally, significant details about the model and experimentation process are missing (how large is it? How long does it take to train and what are the training dynamics?).

Several other works related to optimization of ML workloads and ML for CO are not cited:
- [Neural Combinatorial Optimization with Reinforcement Learning](https://research.google/pubs/pub45821/)
- [Learning Combinatorial Optimization Algorithms over Graphs](https://arxiv.org/abs/1704.01665)
- [Value Learning for Throughput Optimization of Deep Learning Workloads](https://proceedings.mlsys.org/paper/2021/hash/73278a4a86960eeb576a8fd4c9ec6997-Abstract.html)

---

> ### Author Response · Authors · 2022-08-02
> **Response to Reviewer bMHc (Part 1)**
>
> We thank Reviewer bMHc for the constructive feedback. We try to address your concerns below:
>
> 1. **The paper lacks ablations on some of the architectural contributions – some decisions seem somewhat arbitrary as a result. As an example: why is an additional node-wise MLP layer added to follow MHA sub-layers in the model? Is this layer a prerequisite to training, or is it simply a change which empirically improved performance?**
>
>     The additional node-wise MLP layer is standard in the transformer architecture [1] and has been considered in other papers solving combinatorial optimization problems using ML [2]. Hence it was included in our architecture. Regarding other architectural choices, we have provided a general response to briefly layout the ablation studies that we conducted and have a detailed analysis in the Supplementary Material.
>
> 2. **The decision to add constituent graphs as other attention heads is sensible, but there is no speculation or study as to why that inductive bias is effective. What other approaches did the authors explore, and why were they successful or not?**
>
>      As discussed in the general response we experimented with other encoder architectures like using MLP, fully connected transformer, GAT and also an attention based GNN by adapting the encoder of [2]. Our results show that all of these encoders are outperformed by Topoformer. MLP and fully connected transformer do not perform well since they do not exploit the graph structure at all which is crucial for graph problems. GAT and the attention based GNN of [2] perform better than the MLP and transformer encoder but are unable to reach the performance level of Topoformer. This is because local message passing along the edges of the graph is not sufficient to solve graph problems which require global graph information [3,4]. The reason why we believe using such constituent graphs is effective, is that it effectively results in message passing on the fully connected graph, but each node can differentiate messages on the basis of its relationship with the other nodes. This can enable the nodes to receive global graph information while retaining an inductive bias of the graph topology. We also did an ablation study of studying a reduced version of Topoformer as discussed in the general response, and find that although it is still better than the other encoder architectures, there is some performance drop compared to using all the constituent graphs in full Topoformer, which validates our architectural choices.
>
> 3. **Details on the generation of the real-world graphs are lacking. Further, the authors do not discuss releasing the set of graphs you used to evaluate your model, which would greatly enhance reproducibility and would further the paper’s contribution.**
>
>     While we agree that revealing the real-world graphs would be valuable, we were not able to secure permission to make these graphs public, as these are proprietary. The reason we included them in the paper was to get confidence that the results that we obtained on the synthetic graphs translate to the real-world graphs, to further cement the effectiveness of our approach.
>     We have nevertheless included a detailed description of how the layered graph dataset is generated in the Supplementary Material, as well as all the hyper-parameters that were used to generate our graphs.
>
> 4. **The discussion of the approaches for generating and the origins of the real-world graphs can be significantly improved to better-understand the differences between the real-world and generated graphs.**
>
>     Please see answer to question 3.
>
> 5. **The authors can do a better job of broadening the treatment of the benefits of the proposed approach to other types of problems in topological ordering. The problem setup and study, while useful, is rather narrow.**
>
>    We have focused on the compiler sequencing problem because it is of key focus in our research given its practical application and it is a crucial piece in the modern compiler workflows to justify the treatment of its own. That said, our method is quite general and is applicable to any problem which includes searching for a topological order over graphs.
>
> 6. **Is it possible this approach could be used to solve other problems in P\# or related complexity classes that have reductions to or from this particular constrained topological ordering problem? Additional discussion of this might be compelling.**
>
>    Please see answer to question 5.

---

> ### Author Response · Authors · 2022-08-02
> **Response to Reviewer bMHc (Part 2)**
>
> 7. **Finally, significant details about the model and experimentation process are missing (how large is it? How long does it take to train and what are the training dynamics?)**
>
>    We have provided the details about the model architecture in the Supplementary Material. Let us recall them below for convenience.
>
>     *Model architecture* - We use Topoformer with number of layers $n_{layers} = 4$, embedding dimension $d = 256$, number of heads $n_{heads} = 10$ for each MHA operation on the seven graphs listed in section 4.1 and the query and value dimension of $64$ for each head of MHA. The MLP used in (5) consists of a linear layer ($d_{input} = d_{output} = 256$) with GELU activation followed by another linear layer ($d_{input} = d_{output} = 256$). The MLP used in (7) to generate the node priorities consists of a linear layer ($d_{input} = d_{output} = 256$) with RELU activation followed by another linear layer ($d_{input} = 256, d_{output} = 1$).
>
>     We have also provided some training details in the Supplementary Material, but we nevertheless agree that some, such as the running time and training algorithm, were missing. We have added them to the revised manuscript, and we recall them below as well.
>
>      *Training details*- we trained our model for 326 epochs where in each epoch we provide 1000 training graphs. Also, we provide a new training set in each epoch so that we do not overfit our model on a fixed training set. We found the training to be fairly stable and converged in about 1-2 days.
>
> 8. **Per the above: what is the model size, and how long does it take to train on what scale of hardware? Is training sensitive to any configurations or parameters?**
>
>     Please see answer to question 7.
>
> 9. **Several other works related to optimization of ML workloads and ML for CO are not cited: Neural Combinatorial Optimization with Reinforcement Learning, Learning Combinatorial Optimization Algorithms over Graphs, Value Learning for Throughput Optimization of Deep Learning Workloads**
>
>    We have cited these papers in the revised manuscript.
>
> 10. **Beam search significantly increases the cost of inference for the model; results, however, don’t seem much better. Are there any insights as to why beam search doesn’t yield much improvement?**
>
>     Our insight is that, while beam search should intuitively perform better than greedy inference, it is anyway a greedy method of finding the best sequences. As with all greedy methods, is it therefore possible that it exploits local minima by only considering initially promising sequences, instead of exploring less promising ones which might ultimately lead to cost improvements in the longer run. It is reasonable to expect such behavior in the case of relatively small beam sizes (such as the one that we use in the paper), and in the case of cost distributions which are strongly peaked and multimodal.
>     In [3], which we cite, it is pointed out that in the case of the TSP, a policy trained via REINFORCE ends up anyway being quite peaked, confident about its decisions. Moreover, the way that our synthetic graphs are constructed (by fixing the memory cost of all nodes in a layer to the same value) appears likely to produce degeneracy (i.e. many sequences with the same cost) and multimodality in the cost distribution induced by the learned policy.
>     We surmise that these three contributions conspire to induce the counter-intuitive behavior of beam search reported in table 1.
>
> 11. **Have the authors verified that the orderings produced by the model work in practice? A real experiment using the constructed computation graphs would be very compelling.**
>
>     We fully agree,  and we are currently working on that.
>
>
> References:
>
> [1] [A. Vaswani, N. Shazeer, N. Parmar, J. Uszkoreit, L. Jones, A. N. Gomez, L. Kaiser, and I. Polosukhin, “Attention is all you need,” *Advances in neural information processing systems, vol. 30, 2017*](https://proceedings.neurips.cc/paper/2017/hash/3f5ee243547dee91fbd053c1c4a845aa-Abstract.html)
>
> [2] [W. Kool, H. van Hoof, and M. Welling, “Attention, learn to solve routing
> problems!” in *International Conference on Learning Representations, 2019*](https://arxiv.org/abs/1803.08475)
>
> [3] [C. K. Joshi, Q. Cappart, L.-M. Rousseau, and T. Laurent, “Learning TSP
> requires rethinking generalization,” in *International Conference on Princi-
> ples and Practice of Constraint Programming, 2021*](https://arxiv.org/abs/2006.07054)
>
> [4] [L. Xin, W. Song, Z. Cao, and J. Zhang, “NeuroLKH: Combining deep
> learning model with lin-kernighan-helsgaun heuristic for solving the traveling
> salesman problem,” in *Advances in Neural Information Processing Systems,
> 2021*](https://proceedings.neurips.cc/paper/2021/hash/3d863b367aa379f71c7afc0c9cdca41d-Abstract.html)

---

### Author Response · Authors · 2022-08-02
**General response to reviewers - ablation studies**

We would like to thank the reviewers for their thoughtful comments and efforts towards improving our manuscript. Reviewers asked for ablation studies on the effectiveness of different choices in our architecture and comparison with neural baselines.
We would like to point out that we had ablation studies and comparison with neural baselines in the supplementary material, but it was not pointed out explicitly in the main text. We have also added some new ablation studies based on Reviewer BLAK's comments. We briefly describe those studies below:

 **Encoder ablation** -  For the encoder ablation we try four different architectures: MLP, fully connected Transformer, GAT and a reduced version of our Topoformer. We keep the decoder fixed to our non-autoregressive decoder for these experiments. We find that the performance of both *MLP* and *Transformer* is significantly worse than our *Topoformer* encoder. Moreover, the vanilla version of GAT (*GAT forward only*), which does message passing only on the edges of DAG, is also outperformed by our Topoformer architecture. We also consider GAT encoder which does additional message passing on the reverse edges of the DAG and refer to this setting as *GAT forward+backward*. This improves the performance of GAT but its mean performance is still worse than Topoformer's.
In the reduced version of our Topoformer architecture, we only do message passing on the edges and corresponding backward edges of the input DAG. We find that using the full Topoformer (message passing on all the seven graphs listed in section 4.1) leads to better performance than *Topoformer with forward+backward message passing*. This shows the benefit of global message passing between all the nodes which is enabled by the full Topoformer.

We provide the average \% gap from approximate DP for different encoder choices in greedy inference mode below:
| Algorithm      | \% gap (500 node graphs) | \% gap (1k node graphs) |
| :-- | ----------- | ------------|
| MLP      | 8.31       | 2.95|
| Transformer   | 8.46   | 3.09|
| GAT (forward only) | 5.94 | 1.33 |
| GAT (forward+backward)| 4.84 | 0.90|
| Topoformer (forward+backward)| 4.82 | 0.76 |
| Full Topoformer | 4.31 | 0.47 |

Please see the supplementary material for the complete table and a more comprehensive description of the experiments.

**Decoder ablation** - Here we study the effect of using an autoregressive decoder of [1] coupled with Topoformer encoder. We observe that the run time of our decoder is almost 2x faster than the auto-regressive decoder on 1k node graphs while achieving similar performance in terms of peak memory.


**Encoder+Decoder ablation** - Finally we also carried out an experiment by adapting the encoder-decoder architecture of [1] to our problem. The encoder of [1] is an attention based GNN while the decoder is an autoregressive decoder. We find that our model outperforms the model of [1] by a significant margin.

We took care to more clearly refer to these ablations in the main body of the paper for better readability. Also, all the changes in the manuscript have been highlighted in red for clarity.

References:

[1] [W. Kool, H. van Hoof, and M. Welling, “Attention, learn to solve routing problems!”
in *International Conference on Learning Representations*, 2019.](https://arxiv.org/pdf/1803.08475.pdf)

---

### Meta-Review · Area_Chair_7SRM · 2022-08-26

**Recommendation:** Accept
**Confidence:** Certain

**Metareview:**

The reviewers unanimously considered that this paper should be accepted for publication.  Among the strengths noted are is the novelty and relevance of the method, the experimental results and finally it was considered to be well-written.


**Award:**

No

---

### Decision · Program_Chairs · 2022-09-14

Accept